

# Extent, patterns and demographic correlates for physical fighting among school-attending adolescents in Namibia: examination of the 2013 Global School-based Health Survey

Laura E. Davis[1,2], Anne Abio[3], Michael Lowery Wilson[3,4] and Masood Ali Shaikh[3]

[1] Center for Injury Prevention and Community Safety, Peercorps Trust Fund, Dar es Salaam, Tanzania
[2] Department of Epidemiology, Biostatistics and Occupational Health, McGill University, Montreal, Quebec, Canada
[3] Injury Epidemiology and Prevention Research Group, Turku Brain Injury Center, Division of Clinical Neurosciences, University of Turku, Turku, Finland
[4] Heidelberg Institute of Global Health, University of Heidelberg, Heidelberg, Germany

Corresponding author
Michael Lowery Wilson,
michael.wilson@uni-heidelberg.de

## ABSTRACT

**Background**. Physical fighting is particularly detrimental for young people, often affecting other areas of their developing lives, such as relationships with friends and family and participating in risky behaviors. We aim to quantify the amount of problematic physical fighting in Namibian adolescents and identify modifiable risk factors for intervention.

**Methods**. We used the Namibia 2013 Global School-based Student Health Survey (GSHS). This survey collects health-related information on school-attending adolescents in grades 7 to 12. We defined physical fighting as having participated in at least two physical fights in the 12 months prior to responding to the survey. Factors that may be associated with physical fighting were identified a prior based on the literature and included age, sex, anxiety, suicide planning, truancy, physical activity, bullying victimization, presence of supportive parental figures, presence of helpful peers, extent of social network, and food insecurity. Multivariable logistic regression models were created to identify factors associated with physical fighting.

**Results**. A total of 4,510 adolescents were included in the study. A total of 52.7% female. 16.9% of adolescents reported engaging in at least two physical fights in the previous year. Factors associated with an increased odds of physical fighting included having a suicide plan, anxiety, truancy, food deprivation and being bullied. Increased age and loneliness were associated with a decreased odds of physical fighting.

**Conclusion**. This study identifies problematic physical fighting among adolescents in Namibia. We recommend public health and school-based programming that simultaneously targets risk behaviours and conflict resolution to reduce rates of physical fighting.

## INTRODUCTION

Physical fighting is consistently associated with serious injury and among adolescents is particularly problematic as it is strongly associated with both bullying (perpetration and victimization), family violence and weakened academic achievement (*Pickett et al., 2002*; *Pickett et al., 2005*). Additionally, individuals who experience physical fighting in adolescence often experience substance use, poor relationships with peers and decreased overall life satisfaction, that extend well into adult life (*Krug et al., 2002*; *Rudatsikira, Muula & Siziya, 2008a*; *Patton et al., 2009*; *Sousa et al., 2010*).

Several risk factors for physical fighting have been identified in adolescents, such as male sex, poor mental health, drug use, bullying and younger age (*Alikasifoglu et al., 2004*; *Rudatsikira, Muula & Siziya, 2008a*; *Pickett et al., 2012*). However, these studies mainly take place in high income countries and quality information regarding the prevalence and general risk factors for physical fighting is lacking in low-income and sub-Saharan African countries. Detailed contextual information on physical fighting risk factors is essential in order to better target adolescents in sub-Saharan African countries who may at particular risk of participating in physical fighting.

In 2004, 51% of adolescents in Namibia reported having been in at least one physical fight that year (*Rudatsikira et al., 2007b*). We aim to provide an updated overview of the prevalence and risk factors for physical fighting among adolescents in Namibia using the Namibia Global School-based Health Survey (GSHS) conducted in 2013. The current study builds upon a previously published study by Rudatsikira et al., where we aim to examine a more contemporary cohort of Namibian adolescents who exhibit physical fighting behavior and include additional risk factors not previously included in the *Rudatsikira et al. (2007b)* study.

## METHODS

### Setting

Namibia is a sub-Saharan African country on the south Atlantic coast. It borders Angola, Zambia, Botswana, and South Africa. It is the driest country in sub-Saharan Africa and has a population of 2.48 million people, with a stable parliamentary democracy. In 2018. approximately 37% of its population was under the age of 14 years (*World Bank, 2019*). Although increasing, Namibia's Human Development Index is low at 0.640 positioning it at 125/188 countries, next to Guatemala (*UNDP, 2016*).

### Sample

We used the publicly available data from Namibia for the Global School-based Student Health Survey (GSHS), 2013 (*CDC, 2013*). The World Health Organization in collaboration with United States Centers for Disease Control developed the methodology for the GSHS. Detailed information on the data collection methods, questionnaire, and procedures may be found elsewhere (http://www.cdc.gov/gshs/). The survey was administered to school attending adolescents and collected self-reported information on indices pertaining to health risk behaviors. In Namibia, 4,531 students in grades 7 to 12 and aged 13–17 (52.7%

female) completed the survey questionnaire; with a 100% school response rate, and an 89% student response rate. We did not exclude any cases, despite 61 records missing sex and 45 records missing age (with 10 records missing both) information; to ensure correct design-based analysis.

## Measurements

The dependent variable, physical fighting, was derived from one question in the GSHS: "*During the past 12 months, how many times were you in a physical fight?*" Response options ranged from "*0 times*", "*1 time*", "*2 or 3 times*", "*4 or 5 times*", "*6 or 7 times*", "*8 or 9 times*", "*10 or 11 times*" or "*12 or more times*". For the purpose of our analyses, participants were classified as having participated in a physical fight if they reported being in two or more fights. If none or one fight was reported, participants were classified as not participating in a physical fight. For 21 records, this information was missing.

We investigated eleven independent variables at the individual level (age, sex, anxiety, suicide planning, truancy, physical activity, bullying victimization, early sexual debut, alcohol use, marijuana use, and cigarette smoking) and four independent variables at the social level (presence of supportive parental figures, presence of helpful peers, extent of social network, and food insecurity). Details on how these variables were created are provided in Table 1.

## Statistical analysis

The distribution of selected independent variables within the dichotomized physical fighting variable was examined first. Differences between physical fighting involvement among the variables were screened for statistical significance using Rao-Scott chi-square tests for categorical variables and the design-adjusted version of $t$-test for continuous variables (age). We then created two sets of survey binary logistic regression models. These were intended to model the ability of the selected independent variables to predict the dichotomized physical fighting variable. The first set of models included only the predictor variable and adjusted for age and sex. The second model included all variables which were significant at a $p$-value of 0.05 at the bivariate level. We reported the measures of association as adjusted (aOR) and unadjusted (OR) odds ratios and associated 95% confidence intervals (CI). All analyses were carried out using Stata 15 (StataCorp, 2017). All proportions—expressed in percentages—are weighted, unless specified otherwise.

## RESULTS

4,531 students in grades 7 to 12 aged 13 to 17 (52.7% female) completed the survey questionnaire. Twenty-one students were excluded with missing 'physical fights' information, resulting in a final sample size of 4,510 adolescents. For comparison purposes we measured 1 or more and 2 more physical fights in the past year, but use 2 or more physical fights as the main outcome throughout the study. Within the recall period, 32.5% (unweighted count: 1,493) of participants reported being involved in 1 or more physical fights and 16.9% (unweighted count: 785) of participants reported being involved in two or more physical fights in the past year, most of whom were male (59.7%). Parents or

**Table 1  Independent variable derivation from GSHS survey data 2013.**

| Survey question | Coding | Variable |
|---|---|---|
| **Individual-level variables** | | |
| How old are you? | 11–18 years (coded continuous) | Age |
| What is your sex? | Female (0) | Sex |
| | Male (1) | |
| During the past 12 months, how often have you been so worried about something that you could not sleep at night? | Most of the time/always (1) | Anxiety |
| | Never/rarely/sometimes (0) | |
| During the past 12 months, did you make a plan about how you would attempt suicide? | No (0) | Suicide Plan |
| | Yes (1) | |
| During the past 12 months, how often have you felt lonely? | Never/rarely/sometimes (0) | Loneliness |
| | Most of the time/always (1) | |
| During the past 30 days, how many days did you miss classes or school without permission? | 0–2 times (0) | Truancy |
| | 3 to or more days (1) | |
| During the past 30 days, on how many days were you bullied? | 0 times (0) | Bullying victimization |
| | 1 or more times (1) | |
| During the past 7 days, on how many days were you physically active for a total of at least 60 min per day? | 3 days or less (0) | Physical activity |
| | 4 days or more (1) | |
| How much time do you spend during a *typical* or usual day sitting and watching television, playing computer games, talking with friends, or doing other sitting activities? | 2 h or less (0) | Sedentary |
| | 3 h or more (1) | |
| How old were you when you had sexual intercourse for the first time? | Never had sex or had after age 14 (0) | Early sexual debut |
| | Had sex at age 14 or earlier (1) | |
| During the past 30 days, on how many days did you have at least one drink containing alcohol? | 0 days (0) | Alcohol use |
| | 1 or more days (1) | |
| During the past 30 days, how many times have you used marijuana (also called dagga, weed, boom, cannibus, stop, grass, pipt, stop, and joint)? | 0 days (0) | Marijuana use |
| | 1 or more days (1) | |
| During the past 30 days, on how many days did you smoke cigarettes? | 0 days (0) | Cigarette smoking |
| | 1 or more days (1) | |
| **Social-level variables** | | |
| During the past 30 days, how often did your parents or guardians understand your problems and worries? | Never/rarely/sometimes (0) | Supportive parental figures |
| | Most of the time/always (1) | |
| During the past 30 days, how often were most of the students in your school kind and helpful? | Never/rarely/sometimes (0) | Helpful peers |
| | Most of the time/Always (1) | |
| How many close friends do you have? | 0 close friends (0) | Close friends |
| | 1 close friends (1) | |
| | 2 close friends (2) | |
| | 3+ close friends (3) | |
| | (coded continuous) | |
| During the past 30 days, how often did you go hungry because there was not enough food in your home? | Never/rarely/sometimes (0) | Food insecurity |
| | Most of the time/always (1) | |

guardians understanding of respondent's problems and worries was reported as either never, rarely, or sometimes by 59.6%. Having made a plan about how one would attempt suicide was reported by 25.5%. Regarding use of addictive substances and having sex; 21.0% reported early sexual debut, defined as having had sex before turning 15 years old.

**Table 2  Distribution of selected factors according to categories of physical fighting among school-attending adolescents in Namibia, GSHS 2013.**

| Variable | Not involved in <2 physical fights ($n = 3{,}725$) | Involved in ≥2 physical fights ($n = 785$) | P-value[a] |
|---|---|---|---|
| Age (SD) | 15.9 (1.8) | 15.7 (1.8) | 0.009 |
| Sex (male) | 44.3 | 59.7 | <0.001 |
| Anxiety | 13.9 | 22.5 | <0.001 |
| Loneliness | 15.1 | 16.7 | 0.041 |
| Food deprivation | 8.2 | 17.9 | <0.001 |
| Close friends (SD) | 1.9 (1.1) | 1.9 (1.1) | 0.54 |
| Bullying victimization | 40 | 67.3 | <0.001 |
| Truancy | 8.1 | 19.2 | <0.001 |
| Physical activity | 26.4 | 28.7 | 0.134 |
| Sedentary | 35.1 | 40.5 | 0.087 |
| Supportive parental figures | 41.1 | 36.7 | 0.062 |
| Helpful peers | 13.4 | 14 | 0.688 |
| Suicide planning | 22.3 | 41.1 | <0.001 |
| Early sexual debut | 18.8 | 32.8 | <0.001 |
| Alcohol use | 32.6 | 38.8 | 0.002 |
| Marijuana use | 3.7 | 13.0 | <0.001 |
| Cigarette smoking | 7.2 | 21.1 | <0.001 |

**Notes.**

All variables are expressed as proportions (in %) with the exception of age and close friends (mean and standard deviation).

[a]Chi-square test for significance.

Alcohol use, marijuana use and smoking cigarette for one or more days, during the past 30 days was reported by 33.6%, 5.3%, and 9.6% respectively.

Table 2 provides the weighted distribution of selected factors according to physical fighting category. In the bivariate analyses, eleven out of the seventeen variables were statistically significantly associated with participants who had been involved in two or more physical fights. The age and sex adjusted analyses (Table 3) for all the variables found to be statistically significant in bivariate analysis, revealed statistically significant associations for all selected variables.

Table 4 provides results of adjusted analysis for all covariates in the model. Compared to those who did not report being involved in physical fighting, those who had been involved in physical fights were males (OR 2.06; CI [1.54–2.76]), more likely to have made suicide plan (OR 2.10; CI [1.66–2.65]), were truant (OR 1.60; CI [1.08–2.38]), food deprived (OR 1.90; CI [1.41–2.57]), having been bullied (OR 2.35; CI [1.79–3.09]), had early sexual debut (OR 1.43; CI [1.20–1.71]), and were cigarette smokers (OR 1.99; CI [1.19–3.32]). While age bestowed protection against involvement in physical fighting (OR 0.89; CI [0.84–0.94]). Anxiety, alcohol use, and marijuana users were not found to be statistically significantly associated with physical fighting. The goodness-of-fit test revealed that this was a good multivariable logistic model for physical fight in Namibian students (F: 0.93,

**Table 3** Multivariate analysis identifying correlates of physical fighting among school-attending adolescents in Namibia, GSHS 2013.

| Main risk factor | OR | 95% CI | *p*-value |
|---|---|---|---|
| Age (SD) | 0.91 | 0.86–0.96 | 0.001 |
| Sex (male) | 1.94 | 1.55–2.42 | <0.001 |
| Anxiety | 1.93 | 1.53–2.45 | <0.001 |
| Food deprivation | 2.45 | 1.81–3.33 | <0.001 |
| Bullying victimization | 3.16 | 2.49–4.00 | <0.001 |
| Truancy | 2.82 | 1.94–4.09 | <0.001 |
| Suicide planning | 2.55 | 2.19–2.96 | <0.001 |
| Early sexual debut | 1.74 | 1.44–2.10 | <0.001 |
| Alcohol use | 1.34 | 1.13–1.59 | 0.001 |
| Marijuana use | 3.62 | 2.31–5.68 | <0.001 |
| Cigarette smoking | 3.14 | 2.38–4.14 | <0.001 |

**Notes.**
Abbreviations: OR, Odds Ratio; 95% CI, 95% Confidence Interval.
All estimates are adjusted for age and sex; age; or sex.

**Table 4** Outcomes of multivariate analysis of variables associated with two or more physical fighting episodes among school-attending adolescents in Namibia, GSHS 2013.

| Variable | Adjusted OR | 95% CI | *P*-value |
|---|---|---|---|
| Age | 0.89 | 0.84–0.94 | <0.001 |
| Sex | 2.06 | 1.54–2.76 | <0.001 |
| Anxiety | 1.37 | 0.97–1.92 | 0.071 |
| Food deprivation | 1.90 | 1.41–2.57 | <0.001 |
| Bullying victimization | 2.35 | 1.79–3.09 | <0.001 |
| Truancy | 1.60 | 1.08–2.38 | 0.002 |
| Suicide planning | 2.10 | 1.66–2.65 | <0.001 |
| Early sexual debut | 1.43 | 1.20–1.71 | <0.001 |
| Alcohol use | 0.96 | 0.75–1.22 | 0.715 |
| Marijuana use | 1.38 | 0.81–2.35 | 0.219 |
| Cigarette smoking | 1.99 | 1.19–3.32 | 0.011 |

**Notes.**
Only those factors found statistically significant in bivariate analysis were used in this model.
Abbreviations: CI, Confidence Interval.
All estimates are adjusted for all variables listed in the table.

*p*-value: 0.5264). With the exception of having been truant and smoked cigarettes, all other covariates that were statistically significant at *p*-value of less than 0.05.

## DISCUSSION

This study described the prevalence and factors associated with physical fighting among school-attending adolescents residing in Namibia. We found the prevalence of participating in at least two physical fights among school attending adolescents in Namibia was almost 17%. Factors associated with an increased odds of physical fighting included having a

suicide plan, anxiety, truancy, food deprivation, being bullied, early sexual debut and cigarette smoking. Increased age was associated with a decreased odds of physical fighting.

Namibia was found to have a relatively low prevalence of physical fighting, especially compared to other sub-Saharan African countries, and even slightly lower when compared to some high-income countries (*Pickett et al., 2012*). For example, a similar survey in Canada found that 19.3% of adolescents participated in two or more fights (*Djerboua, Chen & Davison, 2016*). Interestingly, relative to other sub-Saharan African countries, rates of physical fighting appear to be particularly low in Namibia (*Rudatsikira et al., 2007b*; *Acquah et al., 2014a*). For reference, in Ghana, approximately 32% of adolescents participated in two or more physical fights in the year prior to the survey (*Acquah et al., 2014a*, *Acquah et al., 2014b*). A similar study in Namibia found that 50% of adolescents had been in one physical fight in the last 12 months from the time of the survey, compared to the present study which found only 32.5% (*Rudatsikira et al., 2007b*). This may reflect an improvement in physical fighting between 2007 and 2013, or differences in respondent characteristics between the years.

Cultural or educational differences in Namibia may result in lower rates of physical fighting when compared to other sub-Saharan African countries. Although the human development index in Namibia is low, and poverty as well as income disparity have been shown to be associated with increased physical fighting, there may be other positive influences that discourage physical fighting among adolescents in the country. For instance, school environment has been shown in association with physical fighting among school-aged children (*Larsen, 2003*; *Limbos & Casteel, 2008*). Namibia has free education and spent 9.2% of GDP on education in 2014, perhaps indicating a positive school environment that may mitigate physical fighting in the country (*Ministry of Arts, Culture and Education, 2017*).

We found several factors associated with physical fighting. Physical fighting is predominantly present in young adolescent males; Sex and age have both been established as common risk factors for physical fighting (*Pickett et al., 2005*; *Pickett et al., 2012*). In line with previous literature in low-income countries, we identified the following factors in association with increased physical fighting: having a suicide plan, anxiety, truancy, food deprivation, being bullied, early sexual debut and cigarette smoking (*Rudatsikira, Muula & Siziya, 2008b*; *Rudatsikira et al., 2008*; *Celedonia et al., 2013*). Previous studies have demonstrated clustering of risky behaviors, such as truancy, cigarette smoking and alcohol abuse, which may explain the association of physical fighting with suicide planning and truancy (*Petridou et al., 1997*; *Pickett et al., 2002*; *Acquah et al., 2014a*; *Yang et al., 2017*). Being bullied has been previously shown in association with violence and physical fighting, indicating that those individuals who are victimized are more likely to perpetuate that violence elsewhere (*Rudatsikira et al., 2007a*; *Pickett et al., 2012*). This may also be associated with factors of low income, such as food deprivation. Lower socioeconomic status and income disparities have been found in association with physical fighting in other studies (*Pickett et al., 2012*).

Interestingly, none of the factors we hypothesized to have a positive influence on adolescents, such as supportive parents, positive peers, physical activity and close friends,

were significantly associated with physical fighting on descriptive analyses, although having supportive parents was close (*p*-value = 0.06). In this study, loneliness was associated with decreased odds of physical fighting. One hypothesis may be that adolescents with a smaller social circle are less likely to engage in delinquent behavior such as physical fighting (*Haynie, 2002*).

Although we found relatively low rates of physical fighting in Namibia, this number is still high at almost 17% of adolescents participating in at least two fights and represents an area for intervention. By identifying factors associated with physical fighting policy makers are able to better target populations who may be at particular risk of participating in this risky behavior, such as younger adolescent males and those who may be dealing with other social issues, such as anxiety, suicidal thoughts and bullying. School programming has shown success in reducing physical fighting among school-aged youth and those programs that specifically target high risk youth are particularly successful (*Wilson, Lipsey & Derzon, 2003*; *Espelage et al., 2013*).

This study has several limitations. First, the cross-sectional nature of the survey means that we cannot make any reference to causation, only association. However, this still allows us to identify potential groups at higher risk for physical fighting. Second, the survey does not capture students who were not present on the day of the survey or those students who do not attend school for other reasons. These students may be part of vulnerable populations, such as low-income groups, and may be more likely to express risky behavior, thus potentially underestimating the risk of physical fighting. Third, the survey is self-report which, although anonymous, may result in social desirability response bias, which would bias our results towards the null. Fourth, we were limited to the variables collected by the survey and therefore may be missing important protective factors that may be associated with physical fighting, such as the potential protective effects of family life (e.g., participating in family meals, important discussions with parents and friends) or participating in community activities (*Fontaine et al., 2016*; *Ttofi et al., 2016*; *Vassallo, Edwards & Forrest, 2016*). Finally, we used a robust definition of two or more physical fights in the last year to capture more repetitive behavior, however, this definition makes comparisons to other classifications more difficult. We believe that the associations between risk factors and physical fighting would not change significantly between these definitions, except in strength of association. Despite these limitations, this study provides information on physical fighting in a large updated and representative sample of all school-attending adolescents aged 11-16 in Namibia.

## CONCLUSION

This study provides an updated description of physical fighting in Namibia as well as factors that can be used to identify populations for intervention programs. Although physical fighting was found to be relatively low in Namibia compared to other sub-Saharan countries, it is still a modifiable risky behavior that can be altered in this young population. We recommend public health and school-based programming that simultaneously targets risk behaviors and conflict resolution to reduce rates of physical fighting. Future iterations

of the survey would benefit from the inclusion of more family and peer-related factors which may help to identify additional correlates that may mitigate aggressive behaviors.

### Funding

Michael Lowery Wilson was supported by the Alexander von Humboldt-Stiftung, Bonn, Germany. Anne Abio was supported by the EDCTP/TDR Clinical Research and Development Fellowship Program, World Health Organization, Geneva, Switzerland; a grant from The John Harvey Lowery Foundation, USA; and the University of Turku Joint Research Grant Fund, Finland. The funders had no role in study design, data collection and analysis, decision to publish, or preparation of the manuscript.

### Grant Disclosures

The following grant information was disclosed by the authors:
Alexander von Humboldt-Stiftung, Bonn, Germany.
EDCTP/TDR Clinical Research and Development Fellowship Program, World Health Organization, Geneva, Switzerland.
The John Harvey Lowery Foundation, USA.
University of Turku Joint Research Grant Fund, Finland.

### Competing Interests

The authors declare there are no competing interests. Laura E. Davis conducted research in affiliation with Peercorps Trust Fund but was not compensated financially or otherwise.

### Author Contributions

- Laura E. Davis and Michael Lowery Wilson conceived and designed the experiments, prepared figures and/or tables, authored or reviewed drafts of the paper, and approved the final draft.
- Anne Abio and Michael Lowery Wilson conceived and designed the experiments, authored or reviewed drafts of the paper, and approved the final draft.
- Masood Ali Shaikh conceived and designed the experiments, analyzed the data, prepared figures and/or tables, authored or reviewed drafts of the paper, and approved the final draft.

### Data Availability

The data is available at the CDC: https://www.cdc.gov/gshs/countries/africa/namibia.htm.

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
