# Peer review of "Extent, patterns and demographic correlates for physical fighting among school-attending adolescents in Namibia: examination of the 2013 Global School-based Health Survey"

_PeerJ, doi:10.7717/peerj.9075_

## Round 0.1 · original submission · Major Revisions

· Academic Editor

Major Revisions

This is a very interesting research that explores factors associated with physical fighting in Namibian schools. I agree with the two reviewers' comments and suggestions and would recommend the authors strengthen the introduction and methodology sections. In particular, authors are expected to draw on support from a broader literature to justify the rationale and significance of this research and describe their work in more details. Additionally, in light of the international coverage of PeerJ, discussions on the applicability of findings from this research to similar issues in other countries will help improve the influence of this article.

·

Basic reporting

Abstract:
1) Line 55: Please rephrase this sentence, as all and any physical fighting should be referred to as problematic.
Introduction:

2) Lines 67-71: Please expand this by adding information on morbidity and mortality from physical fighting in adolescents. Also, more recent references should be added.

3) Line 78: Please reconsider the use of phrase ‘’problematic physical fighting’’, as it might suggest that there is one which is not problematic, or precisely define the phrase. Does this refer to 2 or more fights?

4) Introduction fails to clearly indicate the knowledge gap that authors aim to address. Add information on prevalence estimates of physical fighting in the world and in Namibia, if any. Add information on general risk factors for such behavior, and any disagreement in results of previous studies on these risk factors.

5) Lines 76-80: It is not clear how this study builds upon a previous study which was conducted in the US. Please check the reference you used.

Methods:
6) How were missing variables handled?

7) At which level was significance considered in the bivariate model, in order for a variable to be included in the second model?

8) You stated in the abstract, but not in methodology, that factors associated with physical fighting were identified by a literature search. There is no mention of this in the section Methods. Why does the list of independent variables not include factors which literature recognizes as risk factors for physical fighting in adolescents, such as substance abuse, smoking, socioeconomic factors, etc.?

Results:
9) Lines 154-155: The text ‘’as they did not include ‘1’ in their 99% CIs.’’ Is redundant.

Discussion:
10) Lines 173-175: The cited reference is not conducted in Namibia.

11) Lines 192-197: The discussion of risk factors which you found to be statistically significant should be expanded by comparing your results with other studies.

12) Lines 194-197: This sentence needs to be checked, and rephrased, as it states truancy is associated with truancy. Previous studies have showed that increasing number of joint adverse behaviors is associated with increasing odds for physical fighting (‘’Physical fighting and associated factors among adolescents aged 13–15 years in six western Pacific countries’’ 2017, by Yang et al.).

13) Please consider briefly discussing the protective effect of loneliness, as some of the identified risk factors (e.g. being bullied, anxiety, having a suicidal plan) can be associated with loneliness.

14) Lines 229-234: Information on potential risk factors could be missing too.

15) A limitation is also comparability, given the use of different self-reported measures, definitions of variables, etc.

Experimental design

See above.

Validity of the findings

See above.

Additional comments

See above.

·

Basic reporting

The article meets the standard. However could be strengthened, especially the Introduction. It would be strengthened by addressing changes in physical fighting between the two survey periods.
Line 73, is not true. There is information on the prevalence and general risk factors.

Methods' section
Under setting, describe the age structure of the population as well as the enrolment rates for the study population.

In a few areas, present tense was used instead of past tense. e.g. line 99.."has"
line 110 on definition involvement in a physical fight. The authors chose "2 or more fights" which in some studies "1 or more" was used. Care should be taken when comparing the findings with other findings.

Statistical analysis
what is the benefit of applying Rao-Scott Chi-square test and the design adjusted t test wen the analysis was weighted? Compare the results of these analyses and choose to report more accurate estimates.

Results
line 135: avoid starting sentences with digits
line 147: Results from Table 4 are different from the ones in the paragraph starting in this line
line 152: There is no point to test at 99% confidence level as well.
line 155: no need to give a lecturer on the interpretation of the results of the confidence interval

Discussion
line 173: as the authors stated, the two studies used different definition for physical fighting, hence, it is not worthy to go ahead and make the comparisons.
line 181: write HDI in full.
lines 209-213: this section is repeating paragraph 2 of the Discussion section.
lines 23-219 should be moved to the Introduction as part of the justification of the study.

References
An example of style of referencing from one of your recent articles has the following example:
Albarqouni S, Baur C, Achilles F, Belagiannis V, Demirci S, Navab N. 2016. AggNet: deep learning from crowds for mitosis detection in breast cancer histology images. IEEE Transactions on Medical Imaging 35(5):1313-1321

Clearly if the above format of references is the journals recommended style, then more work is needed to clean up the references. No websites should be indicated.

Experimental design

This is not an experiment.

Validity of the findings

The validity of findings is questionable because the figures in the text do not match those in Table 4.

---

## Round 0.2 · accepted · Accept

· Academic Editor

Accept

In this article, the authors addressed an important issue relevant to educational and health studies and the research conducted is solid that is buttressed by concrete evidence. I want to thank the authors for choosing PeerJ to publish their research and hope to receive your quality research work in the future. At this special, difficult moment caused by the Covid-19 pandemic, we wish you all the best and stay healthy while continuing to be productive in research endeavors.

·

Basic reporting

No comment

Experimental design

No comment

Validity of the findings

No comment

Additional comments

A well argued manuscript